# Toothpaste Composition Effect on Enamel Chromatic and Morphological Characteristics: In Vitro Analysis

**DOI:** 10.3390/ma12162610

**Published:** 2019-08-16

**Authors:** Alexandrina Muntean, Sorina Sava, Ada Gabriela Delean, Ana Maria Mihailescu, Laura Silaghi Dumitrescu, Marioara Moldovan, Dana Gabriela Festila

**Affiliations:** 1Paediatric Dentistry Department 2, Iuliu Hatieganu University of Medicine and Pharmacy, 31 A. Iancu Street, 400083 Cluj-Napoca, Romania; 2Prosthodontics and Dental Materials Department 4, Iuliu Hatieganu University of Medicine and Pharmacy, 15 V. Babes Street, 400012 Cluj-Napoca, Romania; 3Conservative Dentistry Department 2, Iuliu Hatieganu University of Medicine and Pharmacy, 33 Motilor Street, 400001 Cluj-Napoca, Romania; 4Oral Rehabilitation Department 3, Iuliu Hatieganu University of Medicine and Pharmacy, 15 V. Babes Street, 400012 Cluj-Napoca, Romania; 5Department of Polymeric Composites, Babes-Bolyai University, Institute of Chemistry Raluca Ripan, 30 Fantanele Str., 400294 Cluj-Napoca, Romania; 6Orthodontics Department 1, Iuliu Haţieganu University of Medicine and Pharmacy, 31 A. Iancu Street, 400083 Cluj-Napoca, Romania

**Keywords:** toothpaste, enamel colour, fluoride, nano hydroxyapatite

## Abstract

The aim of this in vitro study was to assess the effect of toothpastes, with different compositions, on optical and morphological features of sound and demineralized enamel. We selected twenty-five teeth, recently extracted for orthodontic purposes, for this in vitro study. The teeth were caries free, without stains, fissures, filling or hypoplasia observed at inspection under standard conditions. Teeth were brushed (for 2–3 min, twice a day, for 21 days), with five different toothpastes (four commercially available and an experimental one) containing fluoride and hydroxyapatite. After that, teeth were demineralized with 37% orthophosforic acid (Ultra Etch^®^, Ultradent Products Inc., South Jordan, UT, USA) for 60 s. We repeated the brushing protocol for another 21 days on demineralized enamel. Enamel vestibular surfaces were examined using a spectrophotometer (Vita EasyShade -Vita Zahnfabrik, Bad Sackingen, Germany) and a Scanning Electron Microscope (Inspect S^®^, FEI Company, Hillsboro, OR, USA). Differences were statistically significant for colour parameters L* and ΔE*. SEM evaluation reveals demineralized enamel mineral gain after brushing with selected toothpastes. Toothpastes with specific ingredients can represent a balance between aesthetic and mineralization, and an oral hygiene correct algorithm is able to preserve enamel characteristics during ortodontic treatement with fixed appliances.

## 1. Introduction

The preference for bright white teeth and promotion of dental and facial aesthetics in the media has contributed to an increase in patient interest for healthy and beautiful teeth. Through adolescence, and especially in the course of orthodontic treatment, whitening procedures require caution and for this reason, dental specialists recommended the use of specific dentifrice and accurate brushing technique in order to preserve enamel structural characteristics and appearance [1].

For the duration of orthodontic treatment with fixed appliances, tooth enamel is often at an elevated risk of stains, white spot lesions, caries and dental sensitivity, due to carbonated beverage and food stagnation, facilitated by the appliance and deficient oral hygiene. These factors act on sound enamel and on enamel subjected to adhesives techniques for orthodontic bonding. In line with this, many oral hygiene strategies and products need to address various problems: remineralisation, elimination of dental plaque and preservation of tooth colour [1,2].

Studies showed that the levels of cariogenic biofilm in the oral cavities of orthodontic patients might be 2–3 times higher than in normal individuals experiencing from high rates of biofilm formation [3]. The most widely accepted theory with regard to the role of bacteria in acid production and enamel demineralization is the “ecological hypothesis”, where we define a dental plaque as an active microbial ecosystem in which non-mutans bacteria act for maintaining dynamic stability [4,5]. Dental tissues are continuously undergoing cycles of demineralization and remineralisation. A drop in pH of the oral cavity results in demineralization, which can lead to loss of minerals from the tooth structure. A reversal can occur if pH rises, resulting in deposition of calcium, phosphate, fluoride and other agents [6]. For these reasons, tooth brushing with a multivalent toothpaste looks like the most reliable method not only for plaque control but also for salivary mineral resources acquisition [3].

A toothpaste is defined as a semi-solid material for removing naturally occurring deposits from teeth when used with a toothbrush [3,7]. The tooth brushing procedure involves a mechanical force applied onto the tooth surface over a defined period and determines dental plaque disclosure and the specific action of dentifrice active ingredients [8].

The basic ingredients used in toothpastes (e.g., water, surfactants, thickening agents, flavour) are completed in selected toothpastes composition with remineralising agents and higher amounts of abrasives that are capable of removing or preventing the deposition of stains on the tooth’s surface. Their action consists mainly of removing extrinsic pigmentations without affecting tooth colour, because real whitening action demands the existence of bleaching agents, which consists of free radicals that act on pigments of discoloured teeth [9,10].

The thickness and translucency of the overlying enamel adapt the natural colour of permanent teeth, mainly determined by dentine. The aspect of the teeth whiteness is aesthetically important to many individuals and tooth discoloration is a common patient complaint [3,7]. Tooth colour objective evaluation in dentistry plays an important role in our time, when aesthetic requirements of our patients are higher.

The CIELAB colour system is widely used in dentistry for the determination of colour differences. We described the CIELAB values as “chromaticity coordinates”: L* value refers to ‘lightness’ (a value of 100 corresponds to white and that of zero to black), a* shows red colour on positive values and green colour on negative values, and b* shows yellow colour on positive values and blue colour on negative values [7,10]. Chromaticity coordinates evolution permits an objective evaluation of colour changes, with a better characterization of product action on enamel surface. Human eyes usually more easily perceive changes of L* and b* parameters [11,12].

Colour preservation is important during orthodontic treatment; however, we have to take into account other elements: enamel integrity, dental sensibility and caries risk.

Enamel structure includes both organic and inorganic components. The inorganic component of dental hard tissues consists of biological apatite, Ca_10_ (PO_4_)_6_(OH)_2_. When demineralization of enamel occurs, the enamel hydroxyapatite will dissolve. This is represented with a simplified chemical reaction: Ca_10_ (PO_4_)_6_(OH)_2_ + H^+^ = Ca^2+^ + HPO_4_^2−^ + H_2_O. The left to right direction is demineralization. When calcium (Ca^2+^), phosphate (PO_4_^3−^), and hydroxyl (OH^−^) ions are accumulated, demineralization slows down to the moment when the saliva reaches saturation. When the pH goes up, re-deposition of minerals (remineralisation) will occur and the reaction shifts from right to left [13,14].

Both processes take place on the tooth surface; a substantial number of mineral ions can be lost without destroying enamel integrity but high sensitivity to hot, cold, pressure, and pain can happen. Demineralization is a reversible process, if the partially demineralized hydroxyapatite crystals are exposed to oral environments that favour remineralisation [15].

Various authors have already established the preventive effect of fluoride dentifrices, containing organic and inorganic forms of fluorides. Among the proprieties of fluoride, we notice rapid distribution and greater wettability on enamel surface and highly bacteriostatic and bactericidal effects [16].

In the last decade, nano-hydroxyapatite-containing toothpastes has been introduced showing notable anti-decay activity, protection against hypersensitivity, and preservation of natural translucent whiteness and gloss of teeth. Nano-hydroxyapatite is considered one of the most biocompatible and bioactive materials. Its nanoparticles have similarity to apatite crystals of natural enamel [17].

The aim of this in vitro study was to evaluate the effect of toothpastes with different compositions (commercially available products and an experimental formulation) on natural and demineralised enamel, as regards colour and structural characteristics. The null hypothesis tested is that toothpastes do not produce visible tooth colour changes, and do not determine morphological alterations on the demineralized enamel. 

## 2. Materials and Methods

### 2.1. Sample Selection

This study was conceived under a protocol approved by the University of Medicine and Pharmacy “Iuliu Hatieganu” (Cluj-Napoca, Romania) Ethics Committee (decision nr. 221/17. 05. 2017).

Patients informed consent for orthodontic treatment and used of extracted teeth for scientific purpose was obtained, prior of the beginning of the study. 

Inclusion criteria
Healthy patients (n = 15) aged from 14 to 16 years;The patients present malocclusions with arch-size/tooth-size discrepancies, that have to be solved by premolar extractions (first or second premolars) and a normodivergent skeletal pattern (Figure 1);Patients must have at least 2 premolars indicated for extraction for orthodontic treatment;The teeth were caries free, without stains, fissures, cracks, irregularities, abnormalities hypoplasia or filling observed at inspection, in standard condition. Twenty-five extracted teeth respect these criteria;We selected maxillary upper premolars, because these teeth are the most lightened from the lateral group;We instructed patients, prior to orthodontic treatment, to initiate toothbrushing with upper arch (vestibular surfaces). Topographical positions of upper teeth allow us to presume limited effect of salivary fluid when compared to lower teeth.

### 2.2. Sample Preservation and Preparation

After extraction, the soft tissue and calculus were detached manually using hand scalers, and the organic debris removed with pumice. We stored the teeth at 37 °C in artificial saliva, with a pH of 7.4, contained: 1.5 mmol/L CaCl_2_ (Chempur, Piekary Slaskie, Poland), 50 mmol/L KCl (Chempur, Piekary Slaskie, Poland) and 20 mmol/L tris—tris (hydroxymethyl) aminomethane (Merck, Damstadt, Germany) until used. In order to facilitate handling, roots were embedded in acrylic resin.

### 2.3. Products Employed and Brushing Protocol

Five different toothpastes were used: Lacalut Extra Sensitive^®^ (Theiss Naturwaren, 66424 Homburg, Germany), Lacalut White and Repair^®^ (Theiss Naturwaren, 66424 Homburg, Germany), Biomed Sensitive^®^ (Splat Oral Care, 121099 Moscow, Russia), Aslamed for Sensitive Teeth^®^ (Farmec SA, 400616 Cluj Napoca, Romania) and experimental toothpaste (manufactured by the Polymeric Composites Group, “Raluca Ripan” Chemistry Research Institute, 400294 Cluj-Napoca, Romania). According to manufacturer recommendation, toothpastes respond to various clinical conditions (Table 1)

#### 2.3.1. Treatment 1 (T1)

Teeth were brushing for 2–3 min, twice a day, for 21 consecutive days, with a small amount of toothpaste (the size of a pea) and a manual toothbrush (Oral B^®^ P35 medium toothbrush, Procter & Gamble, Cincinnati, OH, USA) handle by one operator. We employed a manual toothbrush because during orthodontic treatment with fixed appliances patients prefer to make use of this variety of toothbrush. After brushing, teeth were washed with distilled water, and stored at 37 °C in artificial saliva, until used. 

#### 2.3.2. Treatment 2 (T2)

Vestibular enamel surfaces were demineralized with 37% orthophosphoric acid (Ultra Etch^®^, Ultradent Products Inc., South Jordan, UT, USA) for 60 s, and then rinsed and dried for 15 s in order to objectify the demineralized enamel. 

#### 2.3.3. Treatment 3 (T3)

We brushed for another 21 consecutive days, with selected toothpastes, under the conditions and the protocol described above, the teeth that undergoing the demineralization process.

### 2.4. Colour Evaluation

Midpoints of the vestibular surfaces were marked to allow repeated measurements of the same area. The centres of each buccal surface were evaluated with a spectrophotometer (Vita EasyShade^®^ V, Vita Zahnfabrik, Bad Säckingen, Germany), by one examiner, to decrease inter-human variation, with the right angle, according to the Commission Internationale del’ Eclairage (CIE lab) [11]. In order to reduce the effect of external light, colour measurements were made at midday, in the same place, every time. The instrument was automatically calibrated, using an integrated calibration plate, on the base station, after each determination. For every tooth, at each assessment, we recorded three values to allow a better characterisation. The evaluation was made at baseline (T0) and after each treatment (T1, T2, T3).

We calculated differences in parameter changes according to formulas:
ΔL* = L*1 − L*0 (measured values − initial values)
Δa* = a*1 − a*0 (measured values − initial values)
Δb* = b*1 − b*0 (measured values − initial values)

ΔE*ab = [(ΔL*)^2^ + (Δa*)^2^ + (Δb*)^2^]^0.5^ at baseline (T0) and after treatment 1 (T1-T0), 2 (T2-T1) and 3 (T3-T1) respectively. Values that are the clinically acceptable for colour changing and perceived by human eye are around 3.3 for ΔE* and 2 for ΔL* [11,12].

### 2.5. Morphological Characterization

Scanning Electron Microscopy (SEM) observations were carried out by means of Inspect S^®^ Scanning Electron Microscope (FEI Company, Hillsboro, OR, USA) to assess morphological changes. Enamel surfaces were examined by one examiner, in order to reduce human variation, at X50 and X300 magnification.

### 2.6. Statistical Analysis

All collected data were statistically analysed using IBM SPSS (Windows, Version 20.0, IBM Corp. Armonk, NY, USA). Repeated-measures analysis of variance (ANOVA) was performed to analyse colorimeter parameters (colour alteration (ΔE*), luminosity (ΔL*), alteration on the green-red axis (Δa*) and alteration on the blue-yellow axis (Δb*)) with employed toothpaste as a repeated factor. Bonferroni correction was performed. In this study, the statistical significance was set at *p* > 0.05 for all analyses. 

## 3. Results

### 3.1. Toothpaste Composition and Tooth Brushing Effect on Enamel Colour

According to literature, human eye is more sensitive for L*, b* and ΔE* changes [11,12].

#### 3.1.1. Luminosity (L* and ΔL*)

Average values and standard deviations of L* during experimental phases are presented in Figure 2.
The L* value increased after the first brushing protocol T1;Significant differences for L* parameter between toothpastes at T0 and T1 (*p* < 0.05) for lacalut_es, lacalut w_r and biomed_s;After demineralization (T2) we notice significant changes for L* parameter between toothpastes (*p* < 0.05);After demineralization and brushing (T3) L* parameter express values less important compared to T2, but the changes are noticeable when compared to T1.

We obtained important differences for ΔL* parameter especially for lacalut_es, aslamed_st and experim_tp (Figure 3).

#### 3.1.2. b* Parameter

Tooth brushing and toothpastes’ ingredients reveal (on sound and demineralized enamel) variation of yellow–green axis and influence tooth colour perception (Figure 4).
b* parameter present significant differences for the duration of the study, between toothpastes (*p* < 0.05);After T1 significant differences for b*parameter were noticed for lacalut_wr and experim_tp (*p* < 0.05);Biomed_s values for b* were significantly different (T2-T1 and T3-T1) (*p* < 0.05).

#### 3.1.3. ΔE* Parameter

After we complete all the treatments on selected teeth, we notice ΔE* variation, with statistical significant differences (Figure 5).

After demineralization ΔE* values appear higher than 3 for all evaluated toothpastes, and the differences were statistically significant (T2-T1) (*p* < 0.05);After demineralization and brushing, ΔE* values tend to diminish, but the differences remain clinically acceptable and statistically significant (*p* < 0.05);Lacalut w_r differences were higher and statistically significant when compared to lacalut_s and biomed_s.

### 3.2. Toothpaste Composition Effect on Enamel Structure

SEM analyses evaluate modifications in enamel structure, which can affect dental sensitivity and individual caries risk. 

For lacalut_es, SEM images reflect enamel alterations with output on tooth colour and structure (Figure 6).

After the first brushing protocol, sound enamel surface is not completely smooth; pores and superficial irregularities, such as grooves can be observed (d). The demineralization procedure, by ortophosphoric acid 37% for 1 min., removes aprismatic enamel and exposes hydroxyapatite prisms became evident (b). After the second brushing protocol, enamel regain mineral content, via fluoride from toothpaste composition, and crystalline structure was detected (f).

Figure 7 depicts enamel changes after we used lacalut_wr. 

Nano-hydroxyapatite from toothpaste composition restored enamel crystalline structure (c).

Natural ingredients from biomed_s exert a limited protection for enamel structure (Figure 8).

The enamel surface did not present particular changes, like granular structures, after T3.

Fluoride and natural ingredients from aslamed_st ensure mineral gain after demineralization (Figure 9).

The enamel surface is completely smooth (a) and the aprismatic surface layer is uniform (b). SEM evaluation exposed pores and superficial irregularities (c).

Nano-hydroxyapatite from experimental_tp can be an alternative to fluoride to preserve enamel integrity (Figure 10).

The demineralization procedure by 37% ortophosphoric acid, exposed hydroxyapatite prisms (e). After T3, enamel surface became smooth (c, f).

## 4. Discussion

The tooth colour outcomes by their intrinsic colours and the presence of any extrinsic stains [18]. In our study group, we observe greater values for L* parameter, from the beginning of the study (77.76–83.54) and we can assume that this is a consequence of the patients’ young age; the dental enamel did not experience an extensive staining process.

During tooth brushing, a three-phase system is formed by the tooth surface, the toothbrush bristles, and the abrasives between these is responsible for dental plaque and discoloration removal. Dependent on toothpaste composition, tooth brushing may perform not only a mechanical action on the tooth surface but also produce modifications in colour and act as a balance for mineral enamel loss [19,20].

The physical characteristics of the minerals included in toothpaste composition appear to be the major determinant for the mechanical effect, not their quantity or whitening capacity, or rather, their ability to remove enamel surface stains [21]. Our results are in line with studies that show that the use of conventional dentifrices promotes limited changes on enamel colour and appearance that can be perceived by the human eye, but is less important compared to toothpastes containing bleaching products [22,23]. We can conclude that evaluated dentifrices, due to their mechanical action (abrasion), act mainly by removing extrinsic pigmentation, giving a restricted sense of whitening for the sound enamel. Abrasives provide a significant whitening benefit, particularly on smooth surfaces and for these reasons, we presume that the changes are more important in the centre of the buccal surface. On the other hand, abrasive particles are of limited use for areas along the gum line, especially in orthodontic patients, in order to reduce mineral loss in cervical coronal segment (element associated with decay and dental sensitivity) [22,24].

In our study, all toothpastes produced alteration in L*, b* and E* parameters when brushing sound and demineralized enamel. 

For L* parameter variation, it is reasonable to presume that during the brushing protocol, teeth became lighter; changes that would be associated with an increase in perceptual whiteness. A decrease in b* value is most likely to be the result of a reduction in yellowness of the teeth during the demineralization process.

ΔE* parameter variation was greater than 3.3 for the evaluated toothpastes; this feature expresses visible changes for the human eye in enamel appearance. This spreading of the results can be expected as no selections of teeth used in this in vitro study were made on the basis of tooth colour. We can assume that the lighter the teeth, the less the change would be and vice versa, and this element can influence the effect of toothpaste ingredients [25,26,27]. Analysing the composition of dentifrices selected for this study, we observe that they do not contain any substances able to deliver oxygen and subsequent bleaching action; they comprise only high performance abrasives, such as silica and detergents in combination with agents promoting remineralisation. We can assume that a correct brushing technique can augment the potential of evaluated toothpastes to actually influence enamel colour [28,29,30]. 

The experimental model used in the present study for the demineralization of sound enamel was by means of ortophosphoric acid. We used the 37% phosphoric acid in dentistry for etching the enamel and dentine in order to enhance the adhesion of the composite resins. This generates a porous surface, due to the selective dissolution of apatite crystals from the enamel prisms even if the enamel is the most acid-resistant substance in the human body. Apart from this “controlled dental procedure”, the demineralization take place constantly when the environmental acidity (pH) drops below a critical pH level. The main component of enamel is the hydroxyapatite crystal, which is an element of the enamel prism. The space between the columnar prisms is filled with organic components and water, components that are lost when dissolution of the enamel happens [31,32]. 

L* parameter modify after demineralization for all toothpastes, with statistically significant differences (*p* < 0.05). The enamel, due to the prisms’ arrangements, has the ability to transmit light to the underlying dentin, which features several nuances and concede three-dimensional aspects of colour. There is a correlation among the tooth shade and the size of the hydroxyapatite enamel crystals. The demineralization procedure by ortophosphoric acid 37% for 1 min, removes the aprismatic enamel, and hydroxyapatite prisms became evident. From our results, the enamel underwent an augmentation in lightness (differences notable for human eye) when orthophosforic acid act on vestibular surface, a factor that contributes to teeth appearance improvement, performing an artificial sense of whitening, with subsequent mineral loss [33,34]. 

Successive control of caries risk and dental sensitivity of demineralized enamel required the action of the remineralizing elements, in order to restore the structure and ensure enamel mechanical features preservation. Assessed toothpastes have fluoride and/or nano hydroxyapatite that can supplement the minerals dropped after demineralization. These features play an important role during orthodontic treatment, because removing dental plaque from areas around orthodontic attachments is more difficult. We have to take into account that during orthodontic treatment, detached braces re-bonding necessitates an additional adhesive protocol, a supplementary risk factor for enamel surfaces. In line with these, toothpaste is the simplest way to deliver mineral content and act as first barrier against demineralisation. The fluoride and/or hydroxyapatite incorporated in toothpastes composition can compensate mineral loss, but the effect is dependent on patient cooperation; therefore, its outcomes cannot be relied for noncompliant patients. Oral hygiene routine for orthodontic patients with a multivalent product is beneficial for enamel appearance and structure. 

Studies also reveal that whitening dentifrices containing hydrogen peroxide and carbamide may produce lesions on the enamel surface and, for this particular reason; they are used with caution during and after orthodontic treatment or in young patients [35]. 

There are a few studies addressing the clinical efficacy of whitening dentifrices and, for this particular reason, our results emphasize the equilibrium between cosmetic effects and enamel protection when recommending a toothpaste [36,37]. The results of this in vitro study establish the balance between patient apprehensions, associated mainly with dental colour and practitioners’ concerns, related to enamel morphological irreversible alteration [38,39]. Our results decline the null hypothesis that the evaluated toothpastes did not exert effects on enamel optical and structural characteristics. This study design was a short-term study, although the orthodontic treatment period is a long-term treatment, for at least two years. Further research must be done in order to establish the adequate composition for oral hygiene products, capable to limit the unfavourable effect of demineralization on enamel structure and colour [40,41,42].

## 5. Conclusions

Within the limitations of this in vitro study, we assumed that brushing with the evaluated toothpastes has effects on dental enamel appearance. Analysing the ∆E values, we establish a colour alteration that is visible for the patient. Nano-hydroxyapatite and fluoride can ensure a mineral regain for demineralized enamel, a protective factor during orthodontic treatment. It is important to evolve from traditional dentifrice formulations and adopt more biocompatible and bioactive ingredients with multiple actions. Further clinical studies should be performed to evaluate the real effect of constituents used in toothpaste compositions to reach a definite conclusion concerning this matter. 

## Figures and Tables

**Figure 1 materials-12-02610-f001:**
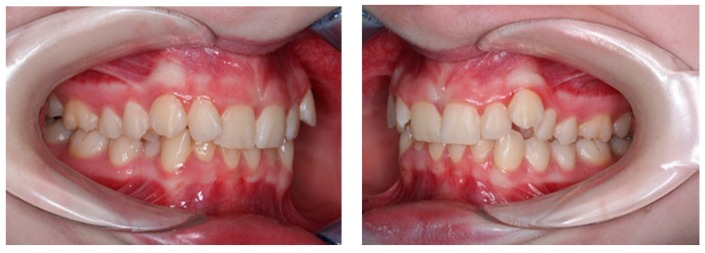
Pre-treatment lateral view of dental arches.

**Figure 2 materials-12-02610-f002:**
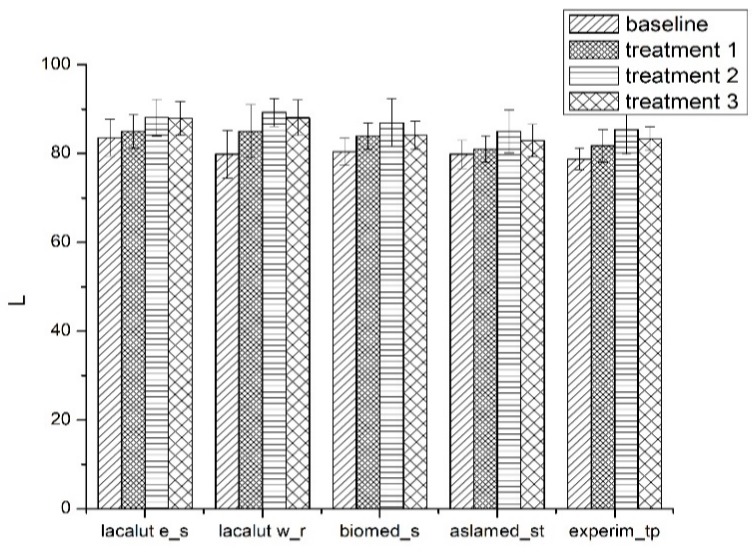
Average values and standard deviations of L* parameter during experimental phases.

**Figure 3 materials-12-02610-f003:**
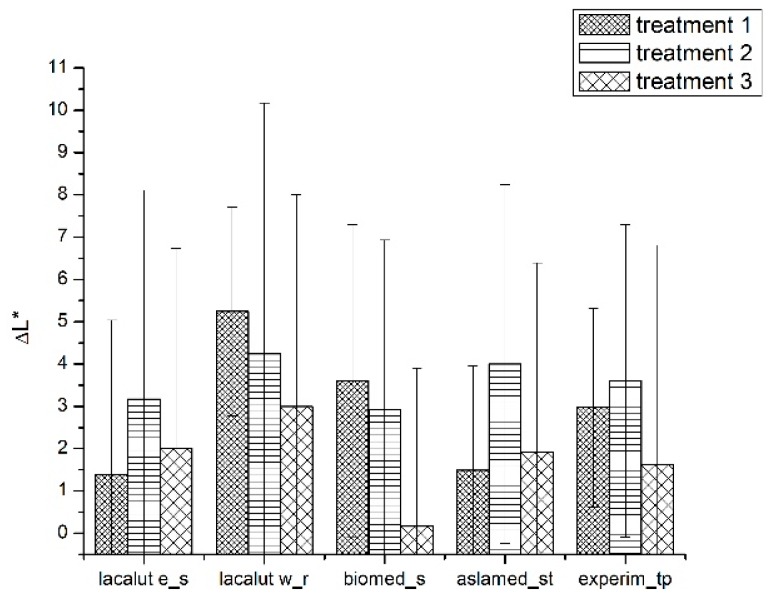
Average values and standard deviations of ΔL* during experimental phases.

**Figure 4 materials-12-02610-f004:**
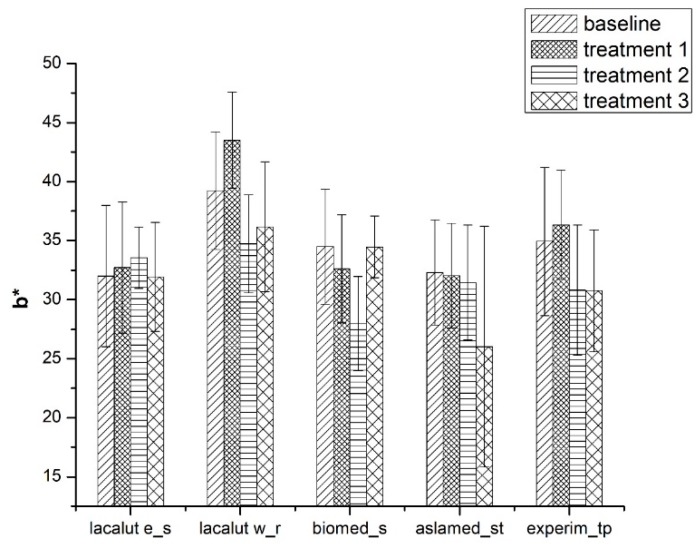
Average values and standard deviations of b* parameter during experimental phases.

**Figure 5 materials-12-02610-f005:**
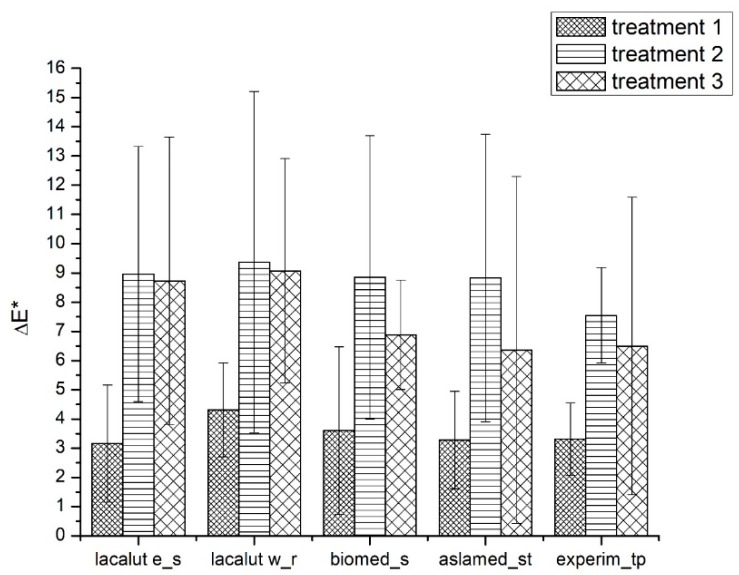
Average values and standard deviations of ΔE* during experimental phases.

**Figure 6 materials-12-02610-f006:**
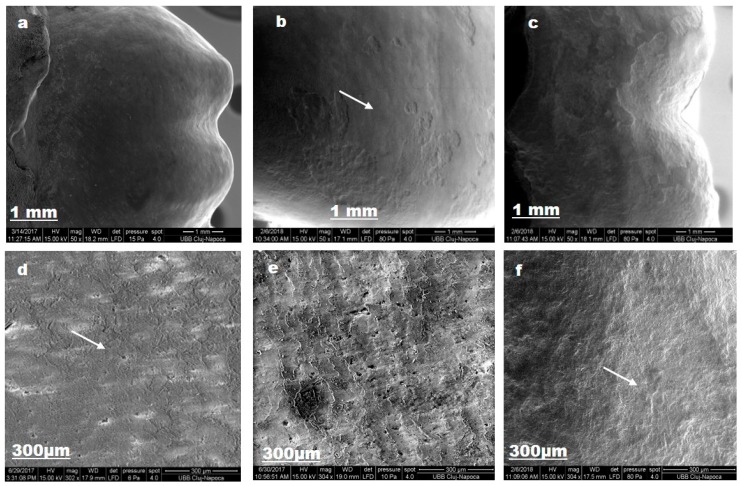
Lacalut_es SEM images for sound enamel after T1 (**a**,**d**), demineralized enamel surface T2 (**b**,**e**), enamel surface after T3 (**c**,**f**).

**Figure 7 materials-12-02610-f007:**
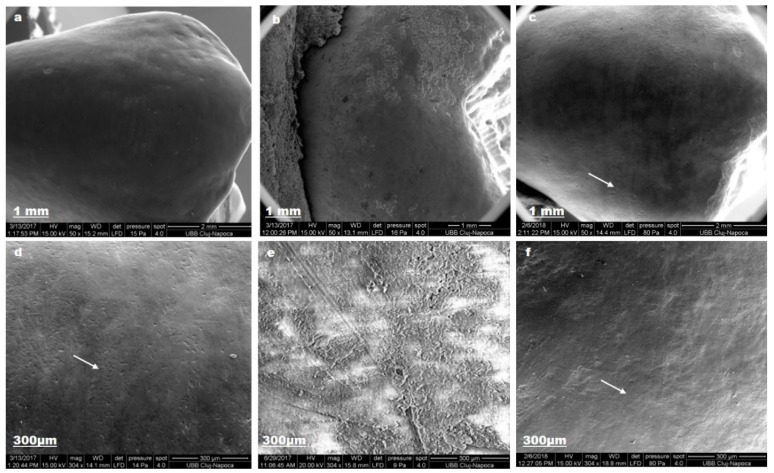
Lacalut_wr SEM images for sound enamel after T1 (**a**,**d**), demineralized enamel surface (**b**,**e**), enamel surface after T3 (**c**,**f**).

**Figure 8 materials-12-02610-f008:**
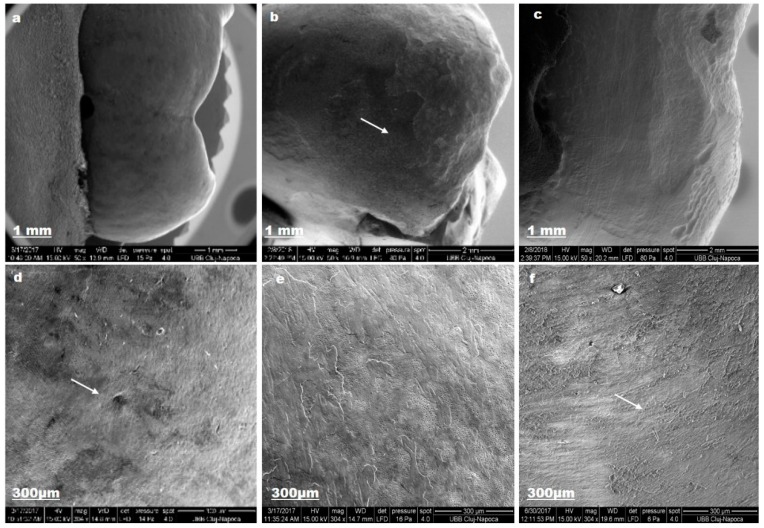
Biomed_s SEM images for sound enamel after T1 (**a**,**d**), demineralized enamel surface (**b**,**e**), enamel surface after T3 (**c**,**f**).

**Figure 9 materials-12-02610-f009:**
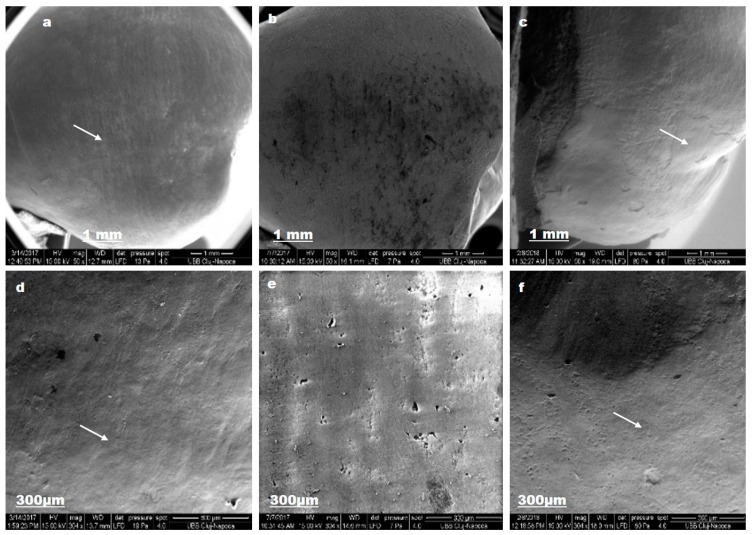
Aslamed_st SEM images for sound enamel after T1 (**a**,**d**), demineralized enamel surface T2 (**b**,**e**), enamel surface after T3 (**c**,**f**).

**Figure 10 materials-12-02610-f010:**
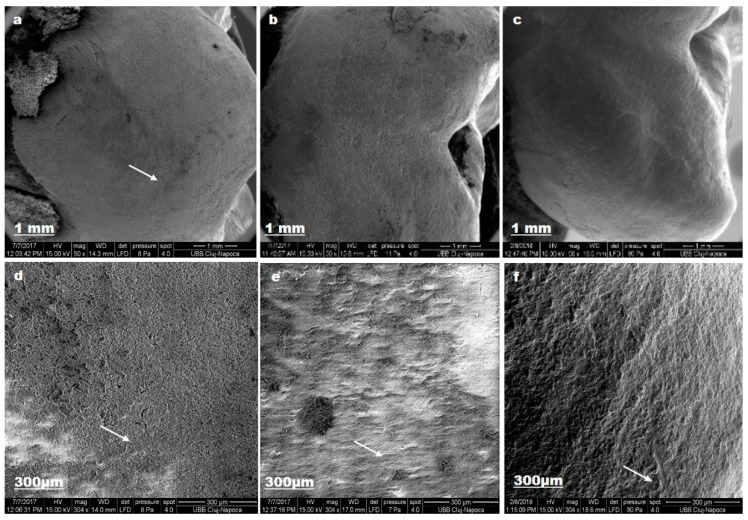
Experim_tp SEM images for sound enamel after T1 (**a**,**d**), demineralized enamel surface T2 (**b**,**e**), enamel surface after T3 (**c**,**f**).

**Table 1 materials-12-02610-t001:** Most importants ingredients and outcomes of commercial and experimental toothpastes.

Toothpaste	Composition	Effect
Lacalut Extra Sensitive (lacalut e_s)	Sodium fluoride, Aluminium salts, Clorhexidine, KCl, silicium dioxide Sodium fluoride Amine	Potassium chloride: improvement of nerve cells Sodium fluoride: caries prevention
Lacalut White and Repair (lacalut w_r)	Hydrated Silica, hydroxyapatite, Pyrophosphate, SLS, Sodium Fluoride(1360 ppm), eugenol	Phosphates: bleach and remove from the surface of the tooth discoloration and sediment Sodium fluoride and hydroxyapatite: remineralisation of enamel
Biomed Sensitive (biomed_s)	L-Arginine, Hydroxyapatite, Natural component (Plantain extract, birch leaf polyphenols and red grape seeds)	Calcium hydroxyapatite: enamel strengthening and eliminating the causes of tooth sensitivity Natural component: dental plaque removal, protection against tooth decay, enamel strengthening
Aslamed for Sensitive Teeth (aslamed_st)	Sodium fluoride, special clay, potassium nitrate, SLS free	Potassium nitrate: clinically proven desensitising effects Special clay: remineralisation of teeth and strengthens their enamel, astringent effect Sodium fluoride: protects against tooth decay Chamomile extract: antimicrobial and anti-inflammatory effect
Experimental toothpaste (experim_tp)	Hydroxyapatite, special clay, potassium nitrate, SLS free	Hydroxyapatite: enamel remineralisation Special clay: remineralisation of teeth and strengthens their enamel, astringent effect

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
