# Peer review of "Toothpaste Composition Effect on Enamel Chromatic and Morphological Characteristics: In Vitro Analysis"

_materials, 2019, doi:10.3390/ma12162610_

Round 1

Reviewer 1 Report

The Manuscript is overall well written and the authors have done a good work considering the enamel remineralisation focusing on the orthodontic patients using various tooth pastes.I just  have one  suggestion to make.The teeth are stored in artificial saliva for further proceedings.There is a well known fact that the PH of saliva will not be same during all the time.There would be changes in the PH values depending on the oral environment and conditions of infections.So it would be valuable to check the effect of these tooth paste on different PH levels in their future studies.

Author Response

Point 1.The Manuscript is overall well written and the authors have done a good work considering the enamel remineralisation focusing on the orthodontic patients using various tooth pastes.I just  have one  suggestion to make.The teeth are stored in artificial saliva for further proceedings.There is a well known fact that the PH of saliva will not be same during all the time.There would be changes in the PH values depending on the oral environment and conditions of infections.So it would be valuable to check the effect of these tooth paste on different PH levels in their future studies.

Response 1.

Thank you very much for your suggestion. Our study will be continued; in order to identify oral cavity microbiology and salivary pH influence on toothpastes actions.

Reviewer 2 Report

The manuscript reported the effect of toothpaste ingredients on the colour and morphological aspects of teeth using 3 steps of treatments, brushing, demineralization by orthophosphoric acid and brushing with selected toothpastes in patients with orthodontic treatments.  The colour change was observed by human eyes of examiners and the morphological change  was measured by scanning electron microscopy. Authors reported brushing with toothpastes effect on enamel apperiance ,and fluoride and nano hydroxyapatite regain minerals for demineralized enamel without hydrogen peroxide and carbamide.

The objectives of this study was so practical for orthodontic patients on aeshethics and prevention of hypersensitivity and dental caries. The study methods in this study were evaluated appropriate procedures and appliances. The conclusions were derived from the study results.

However, several corrections are required for minor spell check as follws;

 line 209, line 237, line 317-8, line 320-2

Author Response

Point 1. The manuscript reported the effect of toothpaste ingredients on the colour and morphological aspects of teeth using 3 steps of treatments, brushing, demineralization by orthophosphoric acid and brushing with selected toothpastes in patients with orthodontic treatments.  The colour change was observed by human eyes of examiners and the morphological change  was measured by scanning electron microscopy. Authors reported brushing with toothpastes effect on enamel apperiance ,and fluoride and nano hydroxyapatite regain minerals for demineralized enamel without hydrogen peroxide and carbamide.

The objectives of this study was so practical for orthodontic patients on aeshethics and prevention of hypersensitivity and dental caries. The study methods in this study were evaluated appropriate procedures and appliances. The conclusions were derived from the study results.

However, several corrections are required for minor spell check as follws;

 line 209, line 237, line 317-8, line 320-2

Response 1.

Thank you very much for your suggestion. We made the corrections and we used track changes.

Reviewer 3 Report

The paper is well structured and easy to be read. The topic is interesting and actual, overall the paper has merit.

Just one minor concerns is about the short number of the investigated teeth. 25 teeth for evaluating 5 different toothpastes maybe it is to low. 

It should be really important if authors could enlarge the study data with a number about 100 teeth in order to have more significant data about the study

In the Sample selection authors stated "....from patients aged from 14 to 16 years were used for this 114 in vitro study. The teeth were recently extracted, caries free, without stains, fissures, filling or 115 hypoplasia observed at inspection, in standard condition."

How many patients were involved in the study?

Authors showed the inclusion criteria for the tooth choice and not for the patient choice.

About patients we know that they were involved in orthodontic treatment. More info should be required.

We lower premolar were not included in the study?

Authors showed really impressive image, therefeore clinical image of real size tooth involved in the  study should be performed, in order to have a more complete paper. 

Author Response

Point 1.The paper is well structured and easy to be read. The topic is interesting and actual, overall the paper has merit.

Just one minor concerns is about the short number of the investigated teeth. 25 teeth for evaluating 5 different toothpastes maybe it is to low. 

It should be really important if authors could enlarge the study data with a number about 100 teeth in order to have more significant data about the study

In the Sample selection authors stated "....from patients aged from 14 to 16 years were used for this 114 in vitro study. The teeth were recently extracted, caries free, without stains, fissures, filling or 115 hypoplasia observed at inspection, in standard condition."

How many patients were involved in the study?

Authors showed the inclusion criteria for the tooth choice and not for the patient choice.

About patients we know that they were involved in orthodontic treatment. More info should be required.

We lower premolar were not included in the study?

Authors showed really impressive image, therefeore clinical image of real size tooth involved in the  study should be performed, in order to have a more complete paper. 

Response 1.

Thank you very much for your suggestion.  According to your recommendation we detailed criteria for patients and teeth selection:

Patients informed consent for orthodontic treatment and used of extracted teeth for scientific purpose was obtained, prior of the beginning of the study.

   Inclusion criteria

Healthy patients (n=15) aged from 14 to 16 years; The patients present malocclusions with arch-size/ tooth-size discrepancies that have to be solved by premolar extractions (first or second premolars) and a normodivergent skeletal pattern (figure 1 - added in manuscript); Patients must have at least 2 premolars indicated for extraction for orthodontic treatment; The teeth were caries free, without stains, fissures, cracks, irregularities, abnormalities hypoplasia or filling observed at inspection, in standard condition. Twenty-five extracted teeth respect these criteria; We selected maxillary upper premolars, because these teeth are the most lightened from the lateral group and the patients were instructed, prior to orthodontic treatment, to initiate toothbrush technique with upper arch (vestibular surfaces). Topographical position of upper teeth allow us to presume limited effect of salivary fluid when compared to lower teeth.

Round 2

Reviewer 3 Report

Authors made excellent job addressing all the reviewer requests and note